# An Exploratory Study on the Relation between Companies’ Food Integrity Climate and Employees’ Food Integrity Behavior in Food Businesses

**DOI:** 10.3390/foods11172657

**Published:** 2022-09-01

**Authors:** Waeel Salih Alrobaish, Peter Vlerick, Noëmie Steuperaert, Liesbeth Jacxsens

**Affiliations:** 1Department of Food Technology, Safety and Health, Faculty of Bioscience Engineering, Ghent University, Coupure Links 653, 9000 Ghent, Belgium; 2Department of Work, Organisation and Society, Faculty of Psychology and Educational Sciences, Ghent University, Henri Dunantlaan 2, 9000 Ghent, Belgium

**Keywords:** food integrity, food integrity climate, food integrity behavior, knowledge, motivation, business ethics

## Abstract

Given the need to prevent food fraud within the international food supply chain and the current lack of research on food integrity, in this paper, the relation between the organizational food integrity climate and employees’ food integrity behavior is examined to understand the role of the individual or psychological dimension in food integrity. The construct of food integrity behavior was introduced and defined, and the conceptual model of the food integrity climate in relation to food integrity behavior was elaborated along with study variables and hypotheses. In the proposed model, the potential moderating role of employees’ psychological well-being (i.e., burnout and job stress) was analyzed, and two mediating variables were also proposed (i.e., knowledge and motivation) which both could explain how the prevailing food integrity climate might influence employees’ food integrity behavior. Data was collected through convenience sampling in four Belgian food companies with a total of 118 participating employees through a self-assessment questionnaire. Based on the statistical analysis, it was concluded that a well-developed organizational food integrity climate promotes positive employees’ food integrity behavior. Specifically, results of this semi-quantitative study demonstrated that the companies’ food integrity climate is positively related to the employees’ food integrity behavior both directly and indirectly, and that food integrity knowledge is a partial mediator in the relation between food integrity climate and food integrity behavior, while food integrity motivation is a full mediator. Study limitations and implications are also discussed.

## 1. Introduction

The widespread phenomenon of food fraud represents a current emergency both in terms of financial loss and human health threat [1,2]. Since food fraud (e.g., food adulteration and counterfeiting) is intentionally and deliberately perpetrated by fraudsters for economic purposes, it remains often undetected, since current food safety management systems and technical strategies do not consider the human or psychological dimension within food organizations [3,4]. To tackle food fraud practices, a preventive approach aiming to identify weaknesses that create opportunities for fraudsters needs to be identified, considering both the organizational level (business climate) and the individual level (employee behavior). In this matter, the emerging discipline of food integrity goes beyond food safety, encompassing all the aspects of food processing, handling and monitoring occurring along the food supply chain and considering also the human dimension involved in the actual execution of processes [5,6,7].

In food safety research, a positive relation was demonstrated between a company’s food safety climate and employees’ food safety behavior [8]. However, whether the prevailing food integrity climate [9] in a food organization is associated with employees’ food integrity behavior has not been studied yet. Moreover, it remains unclear through which mechanisms the food integrity climate in a food business may affect the food integrity behavior of its employees and which factors might influence the strength of this potential relation. 

Hence, the aim of this paper is to study the effect of the organizational climate on employees’ behavior in the context of food integrity through an empirical semi-quantitative study. As part of the research objectives, the novel concept of food integrity behavior is introduced and defined along with its components (i.e., compliance, participation and unethical pro-organizational behavior). Next, the potential variables that may influence or explain the relation between food integrity climate and food integrity behavior are identified and explored. Specifically, the potentially moderating or influencing role of employees’ psychosocial well-being (i.e., job stress and burnout) in this relation is analyzed along with the potential mediating or explanatory role of employees’ food integrity motivation and food integrity knowledge. Findings from this study could contribute and serve as a starting point for the development and implementation of food fraud related interventions in the food industry.

## 2. Conceptual Research Model

Figure 1 illustrates the conceptual model of the relation between food integrity climate and food integrity behavior explored in this paper. The specific moderators and mediators which influence or explain this relation are defined, and relative research hypotheses are also presented in the following sections.

### 2.1. Food Integrity Behavior

Current food safety management systems do not specifically include strategies to prevent intentional food contamination and counterfeiting threats. Human behavior and personal characteristics of employees represent a significant underlying cause behind food fraud [4]. In particular, previous research demonstrated that human behavior determines whether food safety and hygiene procedures are followed [8]. We define food integrity behavior as any behavior of an employee which directly or indirectly influences or is related to product, process, people and/or data integrity in a food organization. It includes both observable and unobservable behaviors. We consider three variables to be indicative for the food integrity behavior of an employee. The first two indicators are analogous to the two components Neal and Griffin [10] recognized in their research on physical safety behavior, namely compliance and participation. Moreover, based on the literature review reported below, we identified unethical pro-organizational behavior as the third indicator of food integrity behavior. In the presented research model, food integrity behavior represents the continuous dependent variable (Figure 1).

#### 2.1.1. Compliance

The concept of compliance concerns the extent to which an employee conforms to the ethical standards in the work environment [8]. It represents the legitimate employee behavior that conforms to the values of the company or wider society which are shared by the organization. The notion of food integrity compliance, therefore, encompasses the extent to which employees comply with the organizational standards, values and regulations that potentially affect product, process, people and/or data integrity.

#### 2.1.2. Participation

The idea of participation refers to all voluntary conduct that promotes integrity in the workplace by, for instance, encouraging or helping colleagues [10]. The specific procedures that must be carried out in a food organization are often reported in checklists and step-by-step protocols, but there are also more intangible voluntary behaviors that affect food integrity in a company. The concept of food integrity participation, therefore, encompasses the extent to which employees engage in voluntary behaviors in the organization that have the potential to affect and promote product, process, people and/or data integrity.

The importance of employees’ compliance and participation has already been empirically demonstrated in the context of food safety [8] and radiation safety of patients and employees [11]. However, in the context of food integrity, these constructs have not been explored yet.

#### 2.1.3. Unethical Pro-Organizational Behavior

Since employees’ food integrity behavior may not always involve a positive or desirable conduct (e.g., compliance and participation), we also consider the ability to refrain from engaging in undesirable behavior (i.e., unethical pro-organizational behavior) as another important indicator of food integrity behavior. The notion of unethical behavior is defined as any behavior that is not legitimate and not accepted by society [12]. In the context of organizations, the literature refers to unethical pro-organizational behavior when employees engage in unethical actions, such as lying or withholding information, to benefit the organization or its agents. It is assumed that individuals with strong attachments and identification with their employer may also be the most likely to engage in unethical pro-organizational behaviors, suggesting that employees may do “bad things for good reasons” [13]. The concept of unethical pro-organizational behavior is particularly relevant in the context of food integrity within food organizations, since food integrity tools and strategies consider the intentionality involved in committing food fraud, which is not contemplated in common food safety plans, as food safety hazards occur non-intentionally. Therefore, the idea of unethical pro-organizational behavior in food integrity refers to any employees’ unethical conduct that violates an organizational or corporate code of ethics, norms or values in a food company (e.g., fraudulent or dishonest behavior) at the expense of product, process, people and/or data integrity.

In summary, we define and measure employees’ food integrity behavior in this study as the extent to which employees (1) adhere to norms and values and follow regulations that may affect food integrity in the organization (compliance), (2) voluntarily establish behavior that promotes food integrity in the company (participation) and (3) refrain from unethical pro-organizational behavior. In other words, employees promote food integrity within their organization if they have high compliance and participation and low unethical pro-organizational behavior. Consequently, we study the relation between the food integrity behavior and food integrity climate in food businesses, where food integrity behavior is the dependent variable and food integrity climate is the independent variable (Figure 1).

### 2.2. Food Integrity Climate

Organizational climate is a subjective construct that refers to the shared perceptions and the meaning attached to the company’s policies, practices and values employees experience within the work environment [14]. In particular, food integrity climate refers to how employees perceive food integrity within the organization. As presented in Alrobaish et al. [9], food integrity climate is defined as “the employees’ shared perception of leadership, communication, commitment, risk awareness and resources regarding food integrity within the company’s working environment in terms of product, process, people and data integrity”. To achieve a consolidated food integrity climate, food organizations should consider and strategize all the four elements of food integrity (i.e., product, process, people and data integrity), conceptualized in Manning [5] and explored in Alrobaish et al. [9], within their management systems, defining a specific set of procedures and a code of conduct that will guide employees’ behavior. Since ethical climate is related to ethical behavior [15], in the context of food integrity, food integrity climate (i.e., employees’ perceptions of work environment) may influence food integrity behavior (i.e., employees’ conduct within the organization). In the presented research model, food integrity climate represents the continuous independent variable (Figure 1).

### 2.3. Food Integrity Climate and Food Integrity Behavior

Since food fraud is intentionally perpetrated, it seems relevant to examine whether food integrity climate has an effect on employees’ food integrity behavior within food organizations. According to Sims and Brinkmann [16], integrity and ethical attitudes of leaders may have an impact on employees’ behavior within organizations. The atmosphere that leaders create influences the ethical behavior of employees [13]. Previous research has identified some of the key factors that create opportunities for food fraud (e.g., simplicity of adulteration and counterfeiting, availability of technology and knowledge to adulterate, accessibility to materials in production, transparency of supply chain network) [4]. Despite the fact that preventive and control strategies are being developed to control and reduce food fraud on a global and organizational level, these do not yet consider the individual level [5]. Previous studies have shown that employees’ behavior is also shaped by the organizational climate in the context of food safety [8] and radiation safety of patients and employees [11]. However, in the context of food integrity, this relation has yet to be explored. Accordingly, we expect that employees perceiving a positive food integrity climate within their food organization will likely behave ethically in terms of food integrity (product, process, people and data integrity). Therefore, we formulate the following hypothesis that we aim to test in this study:

**Hypothesis** **1.**
*Food integrity climate is positively related to food integrity behavior.*


### 2.4. Moderators

A moderator variable affects the strength and direction of the relation between two variables. In this study, job stress and burnout are proposed as indicators for psychological well-being and potential factors moderating the relation between food integrity climate and employees’ food integrity behavior. Workers experiencing high levels of stress and burnout symptoms are a reality for many organizations, according to research in the food and beverage industry [17]. Across different sectors, it was revealed that one out of three employees face high work demands and high work intensity [18]. Financial costs for organizations are, in turn, high because stress leads to absenteeism, accidents, healthcare expenses, employee turnover and declining productivity [19,20]. If the positive relation between organizational climate and employee behavior is weakened by high levels of job stress and burnout, organizations will have to manage the psychosocial well-being of their employees to prevent negative outcomes (e.g., food fraud and unethical work conduct) and to promote positive outcomes (e.g., food integrity and ethical work conduct). Therefore, we aim to examine whether the relation between companies’ food integrity climate and employees’ food integrity behavior is influenced by employees’ psychological well-being. In particular, we expect that the positive relation between food integrity climate and food integrity behavior is weakened by lowered psychological well-being (i.e., high job stress and high burnout), implying that this relation will be stronger in employees with high psychological well-being compared to those with low psychological well-being. In the presented research model, job stress and burnout represent continuous independent variables (Figure 1).

#### 2.4.1. Job Stress

Job stress is defined in accordance with the Effort–Reward Imbalance (ERI) Model of Siegrist [21] as the degree of (im)balance between experienced job requirements or demands and the perceived rewards at work. If employees work hard (e.g., high effort) and do not get enough in return (e.g., low rewards), they will likely experience stress and arousal, which will possibly lead to physical and psychological complaints. In the long run, this imbalance creates an external pressure on employees, which can cause adverse outcomes such as tension, strain, nervousness, reduced physical and mental health and poor sleep quality [22,23]. Besides these direct consequences of job stress, there are also more latent consequences that may occur, such as workplace problems, disrupted relationships and conflicts among employees or between leaders and operators, employee turnover, negative corporate image, reduced productivity and poor quality of the end products [24,25]. If the quality of the final products is affected due to employees’ stress, this might become detrimental for the integrity of the products delivered to consumers and even for public health. Previous research in non-food organizations showed that the safety climate outcomes are influenced by employees’ psychological well-being and the work conditions in which they are employed [26]. Since job stress and its related negative personal and work effects are omnipresent and largely documented in a variety of industrial sectors [20,27] and professional groups [28,29], we assume that job stress might affect the relation between food integrity climate and employees’ food integrity behavior. Therefore, we formulate the following hypothesis that we aim to test in this study:

**Hypothesis** **2A.**
*Job stress moderates the relation between food integrity climate and food integrity behavior.*


#### 2.4.2. Burnout

Burnout is defined as a prolonged response to chronic emotional and interpersonal stress at work and consists of three core dimensions: emotional exhaustion, depersonalization and diminished personal accomplishment [30,31]. Depersonalization is characterized by a distant and indifferent attitude towards work. In response to the prolonged experience of work stress, employees may cognitively distance themselves from their job and develop a cynical attitude [32]. In the context of food organizations, it can be argued that this may be detrimental for food integrity behavior and could increases the risk of food fraud. By experiencing an imbalance between the effort spent and the reward received, employees may also have feelings of inefficiency and incompetence [31], and, as a result, they might become less motivated to act with integrity. Similarly, workers who have high levels of exhaustion (on physical, cognitive and emotional levels) might experience a reduced energy while working, which may hinder them from committing to food integrity [33]. Since burnout is prevalent within the food industry, this variable was considered as a possible moderator in the relation between food integrity climate and employees’ food integrity behavior. Hence, we formulate the following hypothesis that we aim to test in this study:

**Hypothesis** **2B.**
*Burnout moderates the relation between food integrity climate and food integrity behavior.*


### 2.5. Mediators

A mediator variable explains the process through which two variables are related. Two potential mediators in the relation between food integrity climate and employees’ food integrity behavior in food companies are tested in this study, namely knowledge regarding food integrity (i.e., food integrity knowledge) and motivation to act with integrity (i.e., food integrity motivation). Previous research has, in fact, demonstrated that the organizational climate does not only have a direct effect, as formulated above in Hypothesis 1, but also an indirect effect on employee behavior. For instance, in the food industry, it appears that food safety climate in a food company has both a direct (without mediation) and an indirect effect (mediated by motivation and knowledge) on employees’ food safety behavior [8]. In the hospital sector, it was shown that the radiation safety climate has both direct and indirect effects (partly mediated by knowledge, but not by motivation) on the radiation safety behavior of the surgical staff in the operating room [11]. Therefore, and applied to food integrity, we aim to examine whether the relation between companies’ food integrity climate and employees’ food integrity behavior could be explained by employees’ food integrity knowledge and motivation. In particular, we expect that the positive relation between food integrity climate and food integrity behavior is mediated by both variables, implying that food integrity climate might foster food integrity knowledge and food integrity motivation among employees, which, in turn, may stimulate good food integrity behavior. In the presented research model, food integrity knowledge and motivation represent continuous independent variables (Figure 1).

#### 2.5.1. Food Integrity Knowledge

Food integrity knowledge is defined as the degree of knowledge an employee possesses about the company’s ethical standards, values, principles, rules and procedures about product, process, people and data integrity in the organization. Since knowledge in the context of physical security [34], food safety [8] and radiation safety [11] was proven to play a mediating role between the prevailing organizational climate and employee behavior, we assume that the relation between the food integrity climate in a food company and the extent to which employees behave with integrity can be explained by an increased knowledge about food integrity (in terms of product, process, people and data integrity). In fact, organizational climate might influence employees’ knowledge, but also knowledge itself might influence human behavior through increased exchange of information about integrity, both formally (e.g., in meetings and through training) and informally (e.g., during informal discussions in the workplace) [8]. Accordingly, we formulate the following hypothesis that we aim to test in this study:

**Hypothesis** **3A.**
*Food integrity knowledge mediates the relation between food integrity climate and food integrity behavior.*


#### 2.5.2. Food Integrity Motivation

Food integrity motivation is defined as the willingness of an employee to commit to food integrity within the organization. Since it was proven that motivation in the context of physical safety [34] and food safety [8] plays a mediating role between the prevailing organizational climate and employee behavior, we assume that the relation between food integrity climate in a food company and the extent to which employees behave with integrity can be explained by an increased level of motivation to act with integrity (in terms of product, process, people and data integrity). In fact, organizational climate might influence employees’ motivation, but also motivation itself might shape employees’ behavior by augmenting, for instance, the desire to imitate or reciprocate leaders’ integrity actions [8]. Accordingly, we formulate the following hypothesis that we aim to test in this study:

**Hypothesis** **3B.**
*Food integrity motivation mediates the relation between food integrity climate and food integrity behavior.*


## 3. Materials and Methods

### 3.1. Measurement Instruments

To measure the study variables and test the hypotheses formulated above, a self-assessment questionnaire consisting of a total of 61 questions was developed based on a literature study and discussion with subject-matter experts with expertise on food safety management as well as work and occupational health psychology. The English version of the questionnaire is reported in Appendix A.

#### 3.1.1. Food Integrity Climate Assessment

To measure the food integrity climate of the four participating companies through the perceptions of their employees, the food integrity climate (FIC) self-assessment tool, developed and validated in Alrobaish et al. [9], was used. The FIC tool allows food organizations to gain a deep insight on both the technical and managerial aspects as well as the human dimension behind the company’s food integrity, through the assessment of 20 indicators obtained by combining five key organizational climate components (i.e., leadership, communication, commitment, risk awareness and resources) [35] with four food integrity elements (i.e., product, process, people and data integrity) [5]. An example of a FIC tool statement is: “In my company, leaders and employees act properly and constructively to solve issues that affect process integrity (e.g., leaders are prepared to face emergencies; employees are ready to correct incidents or non-compliances on the production line)”. Respondents were asked to self-evaluate each of the twenty statements based on a five-point Likert answer scale, ranging from “strongly disagree” (1) to “strongly agree” (5), where responses closer to five imply a higher perceived food integrity climate. In each of the four food integrity elements sections, one item was formulated negatively to increase the accurateness and reliability of responses [36], hence, these four scores were reversed for analysis. The reliability and internal consistency of the FIC tool is high (Cronbach’s Alpha value = 0.89) [9].

#### 3.1.2. Food Integrity Behavior Assessment

Food integrity behavior was measured by means of fourteen items, each to be scored on a five-point Likert response scale from “strongly disagree” (1) to “strongly agree” (5). In line with our hypotheses, we distinguished three subscales: food integrity compliance (five items), food integrity participation (five items) and unethical pro-organizational behavior (four items). An example item of the food integrity behavior tool is “I voluntarily perform additional tasks and activities within my organization that benefit the well-being or health of the employees in my company”. For these three subscales, the higher was the score given by the participants, the better was their self-reported food integrity behavior. As compliance and participation items were formulated positively and the unethical pro-organization behavior items negatively, the scores for unethical pro-organizational behavior were reversed for analysis. The reliability of the food integrity behavior tool is high (Cronbach’s Alpha = 0.84).

#### 3.1.3. Job Stress Assessment

The measurement of job stress was carried out through a single item, similarly to the procedure adopted by De Boeck et al. [8] in the context of food safety. A seven-point Likert response scale was used, ranging from “never” (0) to “always, daily” (7). The following indicator was applied: “How often do you feel stressed because of your job?”.

#### 3.1.4. Burnout Assessment

Burnout was measured using an abbreviated version of the Maslach Burnout Inventory (MBI) [30], for which the validation was undertaken by Riley, Mohr and Waddimba [37]. For each of the three MBI scales (i.e., emotional exhaustion, depersonalization and diminished personal accomplishment), one item was retained, to be scored by participants on a seven-point Likert response scale from “never” (0) to “always, daily” (7). An example item for the burnout scale is: “I feel mentally exhausted because of my job”. The higher the score given by the participants for the items of this scale was, the higher the level of the self-assessed burnout. The scores of the diminished personal accomplishment item were reversed for analysis since this was formulated positively, unlike the other two items. The Cronbach’s Alpha of the burnout scale is 0.62. Given the limited number of items used in the operationalization (three items) to capture the nomological network of the burnout construct, it was decided not to remove any of the three items.

#### 3.1.5. Food Integrity Knowledge Assessment

In line with the safety knowledge questionnaire by Neal and Griffin [10], six items were developed to measure knowledge related to food integrity. The participants had to answer using a five-point Likert response scale ranging from “completely disagree” (1) to “completely agree” (5). An example item of this tool is: “I am aware of the ethical standards, values and principles that are important to my organization”. The internal consistency of the food integrity knowledge scale is high (Cronbach’s Alpha = 0.82).

#### 3.1.6. Food Integrity Motivation Assessment

In line with the safety motivation questionnaire by Neal and Griffin [10] and the food integrity conceptualization of Manning [5], seven items were developed to measure employees’ motivation to act with integrity in terms of product, process, people and data. Participants were asked to answer on a five-point Likert response scale ranging from “completely disagree” (1) to “completely agree” (5). An example item of this scale is: “I believe that integrity and honesty at work are very important”. The reliability of the food integrity motivation scale is high (Cronbach’s Alpha = 0.83).

#### 3.1.7. Control Variables

Five demographic variables, namely company (A, B, C, D), age, seniority, function (i.e., management, operators daily in contact with food, operators not in direct contact with food) and contract type (i.e., permanent or temporary contract), were included as control variables and assessed through a set of questions at the end of the questionnaire in order to classify participants based on specific employees’ characteristics that may have had an effect on the study variables. These represent the categorical independent variables of the study.

The personality trait conscientiousness was included as the sixth control variable to verify its potential influence on the participants’ estimation of their own food integrity behavior. A person is described as conscientious when this follows standards and rules socially prescribed and tends to be task and goal-oriented at work [38]. Conscientiousness is closely related to integrity. Some authors argue, in fact, that measuring integrity actually means measuring conscientiousness [39]. In this study, this variable was assessed though the Conscientiousness Scale [40] by means of five items. Participants were asked to answer on a five-point Likert response scale ranging from “completely disagree” (1) to “completely agree” (5). An example item of this scale is: “I pay attention to details”. The fifth item of the conscientiousness scale was formulated negatively, therefore, its score was reversed for analysis. The Cronbach’s Alpha of the conscientiousness scale is 0.67. Given the limited number of items used in the operationalization (five items) to capture the nomological network of the conscientiousness construct, it was decided not to remove any of the five items.

### 3.2. Sample Selection

In order to obtain a diversified and heterogeneous sample including small, medium and large organizations, plant or animal-based food companies, operative in different steps of the supply chain (processing, distribution, retail and/or catering), four Belgian food companies were selected with the convenience sampling method based on their various organizational characteristics. A total of 118 participants employed in the 4 different organizations with different demographic characteristics and job functions completed the questionnaire. Regarding the companies’ profiles, Company A is a family-owned food organization with a strong ethical profile, committed to sustainability, producing vegetarian and organic dishes. A total of 22 employees work in Company A, and the response rate of those who completed the questionnaire was 63%. Company B is a family-owned food business with a conservative profile active in the meat processing industry. A total of 59 employees work in Company B and the response rate was 56%. Company C is a food organization producing and distributing raw materials and ingredients for bakeries and catering. Although their company has 300 employees, we have opted, in consultation with the company, to administer the questionnaire only within one of their multiple production areas where 52 employees are employed and where the response rate was 67%. Company D is a food business producing traditional beer with 49 employees. The response rate in Company D was 73%. Regarding the employees’ profiles, participants had a balanced variation in terms of job-demographic characteristics considering their age, seniority and contract type, and, concerning their function within their company, some participants were operators in daily contact with food, some had managerial positions and some were operators not in contact with food (e.g., administration, transportation, technical maintenance). 

### 3.3. Data Collection

A cross-sectional study was executed. Participants individually filled out a paper version of the questionnaire (reported in Appendix A) in the presence of the researchers. Using the back-translation method [41,42], the questionnaire was translated in Dutch and in French, the native languages of the participants. The questionnaire was filled out voluntarily and anonymously to ensure the privacy of the participants. Filling in the questionnaire implied consent, and confidentiality was guaranteed. To ensure a thorough assessment of food integrity climate and food integrity behavior within the participating food companies, the questionnaire was distributed to all employees in the four organizations, including operators in daily contact with food as well as managers and operators who have no direct contact with food.

### 3.4. Data Processing and Analysis

IBM SPSS Statistics version 28 was used to analyze the collected data. A demographic analysis was first performed, providing the percentages of the study participants based on the categories analyzed (i.e., company, age, seniority, function and contract type). Next, a descriptive analysis was conducted, providing the sum scores and mean scores of the constructs assessed (i.e., food integrity climate, food integrity behavior, mediators, moderators and conscientiousness) on the participating sample. The internal consistency of the tools was analyzed, calculating the Cronbach’s Alpha of each scale according to the commonly accepted rule of α > 0.70 [43]. The Pearson’s correlation coefficient was applied to verify bivariate relations between the variables included in the conceptual model of this study (i.e., food integrity climate, food integrity behavior, mediators, moderators and conscientiousness), where correlations with a significance level of *p* < 0.05 were considered. Further, the one-way ANOVA statistic was used to analyze the bivariate relations between the categorical independent variables (i.e., company, age, seniority, function, contract type), including the conscientiousness and the dependent variable (i.e., food integrity behavior), where relations with a significance level of *p* < 0.05 were retained for further analysis. The influence of the independent variable (i.e., food integrity climate) on the dependent variable (i.e., food integrity behavior) was examined by hierarchical linear regression to test Hypothesis 1. The moderation effect of job stress and burnout was analyzed using an interaction term and performing hierarchical linear regression analysis to test Hypotheses 2A and 2B. Finally, the mediation effect of knowledge and motivation was analyzed with a mediation analysis to test Hypothesis 3A and 3B, where the Baron and Kenny [44] procedure was followed. The basic statistical assumptions to perform the mentioned statistical analyses were verified and fulfilled. Statistical results were deemed statistically significant if *p* < 0.05. 

## 4. Results

### 4.1. Descriptive Statistics

#### 4.1.1. Demographics of Respondents

The demographic analysis undertaken over the total sample (n = 118 employees) in the 4 participating food companies highlighted that the majority of the employees are over 41 years of age (61%). With regard to the seniority, participants are equally divided between employees who have worked in their company for less than 10 years (49%) and those who have worked there for over 10 years (51%). Furthermore, 68% of the respondents are in daily contact with food, 10% of the participants have a managerial role and the remaining 22% have no direct contact with food. Finally, the vast majority of the employees in the sample have a permanent contract of indefinite duration (85%), while the remaining 15% work within the organization temporarily (Table 1).

#### 4.1.2. Univariate Results

The study of the continuous variables demonstrated that, on average, employees in the total sample (over all four companies) assigned a relatively high score to their organizations’ food integrity climate (overall mean = 78.08/100) as well as to their food integrity behavior (overall mean = 58.50/70). Moreover, participants rated their overall level of burnout at work as relatively low (overall mean = 8.06/21) and their level of job stress as medium (overall mean = 3.37/7). Finally, employees gave themselves a high score on their perceived level of knowledge about food integrity (overall mean = 25.55/30) and a very high score on their motivation to behave with integrity (overall mean = 30.59/35) (Table 2).

#### 4.1.3. Bivariate Results

The Pearson correlations, calculated to verify the relation between the continuous independent variables and the continuous dependent variable, demonstrated that a significant positive correlation was present between food integrity climate and food integrity behavior (r = 0.54, *p* < 0.01) (Table 2), implying that the higher the employees perceived their company’s food integrity climate, the better they rated their food integrity behavior. 

Moreover, a significant positive relation between the food integrity climate and the mediators’ food integrity knowledge and motivation was observed (respectively r = 0.40, *p* < 0.01; r = 0.58, *p* < 0.01), indicating that a strong food integrity climate is associated with higher food integrity knowledge and motivation. A significant positive association between the mediators and the dependent variable was also observed (respectively r = 0.64, *p* < 0.01; r = 0.71, *p* < 0.01), meaning that the higher the employees assess their own food integrity knowledge and motivation, the better they perceive their food integrity behavior.

On the other side, the moderator burnout was seen to have a significant negative correlation with the dependent variable (r = −0.42, *p* < 0.01). Burnout was also significantly negatively correlated with food integrity knowledge and motivation (respectively r = −0.36, *p* < 0.01; r = −0.37, *p* < 0.01), implying that employees who experience high levels of burnout have lower motivation to behave with integrity and lower levels of knowledge about food integrity. Moreover, job stress, the other moderator in the research model, was not significantly associated with the extent of food integrity behavior (r = −0.04, *p* > 0.05), with food integrity motivation (r = −0.08, *p* > 0.05), nor with food integrity knowledge (r = −0.03, *p* > 0.05).

The continuous control variable conscientiousness showed a significant positive relation with food integrity behavior in the bivariate analysis (r = 0.55, *p* < 0.01), reflecting that the more employees consider themselves conscientious, the better they perceive their food integrity behavior.

To verify the bivariate relation between the five categorical independent control variables and the continuous dependent variable, the one-way ANOVA statistic was used. Results show that only the variable company (A, B, C, D) was significantly associated with the food integrity behavior of employees (F (3, 105) = 5.76, *p* = 0.001). In particular, employees in Company A recorded the highest food integrity behavior (mean = 63.23/70), followed in order by Company D (mean = 59.66/70), Company C (mean = 58.22/70) and Company B with the lowest food integrity behavior (mean = 55.28/70) (Table 1). Remarkably, comparing these food integrity behavior scores per company with the food integrity climate scores per company, the ranking order is identical, with Company A recording the highest food integrity climate score (mean = 84.07/100), followed in order by Company D (mean = 79.97/100), Company C (mean = 76.71/100) and Company B recording the lowest food integrity climate score (mean = 74.77/100). 

The conscientiousness was found to be significantly related to food integrity behavior, also based on the ANOVA analysis (F (10, 97) = 7.40, *p* = < 0.001). On the other side, the extent of food integrity behavior was not significantly related to employees’ age, seniority, function or contract type. Therefore, only company (A, B, C, D) and conscientiousness were retained as control variables for further analyses. 

### 4.2. Hypothesis Testing

#### 4.2.1. Hypothesis 1: Relation between Food Integrity Climate and Food Integrity Behavior

A hierarchical linear regression analysis was performed to verify whether the food integrity climate is positively related to food integrity behavior. In the first step, the control variables (company and conscientiousness) were included. Results show that these two control variables were statistically significantly related with food integrity behavior (F (4, 102) = 19.26, *p* < 0.001), explaining together the 43% incremental variance in the dependent variable. In the second step, results show that the food integrity climate perceived by the employees explained the 8% incremental variance in food integrity behavior (F (1, 101) = 16.50, *p* < 0.001). Thus, a higher score on food integrity climate is associated with a higher score on food integrity behavior (β = 0.31, *p* < 0.001) (Table 3). The positive relation between the perceived food integrity climate and food integrity behavior was confirmed by this hierarchical linear regression analysis, which also corroborates the findings of the bivariate correlations. Therefore, Hypothesis 1 was confirmed. 

Additional hierarchical linear regression analyses were performed per subscale of food integrity behavior (i.e., compliance, participation and unethical pro-organizational behavior). Results show that the food integrity climate was significantly related to the variables compliance (β = 0.39, *p* < 0.001) and participation (β = 0.25, *p* = 0.005), but not to unethical pro-organizational behavior (β = 0.18, *p* = 0.07). Food integrity climate had the largest effect on the subscale compliance, explaining 12% incremental variance in compliance (F (1, 106) = 24.51, *p* < 0.001). Food integrity climate also explained the 5% incremental variance in the subscale participation (F (1, 106) = 8.25, *p* = 0.005). The 3% incremental variance in the food integrity climate in relation to unethical pro-organizational behavior was not significant (F (1, 105) = 3.38, *p* = 0.07) (Table 3). However, conscientiousness was significantly related to unethical pro-organizational behavior (β = 0.27, *p* = 0.01), implying that the more employees consider themselves conscientious, the less they report to engage in unethical pro-organizational behavior at work.

#### 4.2.2. Hypothesis 2A: Job Stress as a Moderator

To verify whether job stress affects the relation between food integrity climate and food integrity behavior, a hierarchical linear regression analysis was performed. In particular, according to Hypothesis 2A, the positive relation between food integrity climate and food integrity behavior should weaken as employees experience high job stress. In a first step, the control variables (company and conscientiousness) were included. As already demonstrated from the testing of Hypothesis 1, the two control variables appeared to be significantly related to food integrity behavior (F (4, 102) = 19.27, *p* < 0.001). In the second step, the variables food integrity climate and job stress were added to test their main effects on food integrity behavior. Results show that these two independent variables explained the 10% incremental variance (F (2, 99) = 10.21, *p* < 0.001). A positive significant main effect of the food integrity climate was present (β = 0.36, *p* < 0.001). In contrast, no significant main effect of job stress was found (β = 0.14, *p* = 0.06). In the third and final step, it was examined whether the interaction effect of food integrity climate and job stress was significant on food integrity behavior. However, the model with this interaction term did not significantly explain an increased variance. Therefore, the interaction effect of food integrity climate and job stress on food integrity behavior could not be confirmed in this study (F (1, 99) = 3.23, *p* = 0.08) (Table 4). Hence, Hypothesis 2A was rejected, since job stress appeared not to have a moderating effect. This implies that the relation between the perceived food integrity climate and food integrity behavior does not differ significantly for employees who may or may not experience high levels of job stress. This also corroborates the findings of the bivariate correlations.

#### 4.2.3. Hypothesis 2B: Burnout as a Moderator

To check whether burnout, in addition to job stress, is a moderator in the relation between food integrity climate and food integrity behavior, a hierarchical regression linear analysis was performed. In particular, according to Hypothesis 2B, the positive relation between food integrity climate and food integrity behavior should weaken as employees experience high burnout. In the first step, the control variables (company and conscientiousness) were included. Again, results show that these two control variables are significantly related to food integrity behavior (F (4, 101) = 18.67, *p* < 0.001). In the second step, the variables food integrity climate and burnout were added as possible main effects on food integrity behavior. Results reveal that these two independent variables explained the 10% incremental variance (F (2, 99) = 1.98, *p* < 0.001). A positive significant main effect of food integrity climate was present (β = 0.27, *p* = 0.001). In contrast, no significant main effect of burnout was found (β = −0.13, *p* = 0.11). In the third step, the interaction effect between food integrity climate and burnout was analyzed. Results show that the interaction effect of food integrity climate and burnout on food integrity behavior was not significant (F (1, 98) = 0.67, *p* = 0.42). Therefore, it could not be concluded that burnout has a moderating effect in the relation between food integrity climate and food integrity behavior due to the lack of a significant interaction effect (β = 0.06, *p* = 0.42) (Table 5). Although, in the bivariate correlations, burnout was seen to have a significant negative correlation with the dependent variable, Hypothesis 2B could not be confirmed by this hierarchical regression linear analysis.

#### 4.2.4. Hypothesis 3A: Food Integrity Knowledge as a Mediator

A mediation analysis was performed following the procedure of Baron and Kenny [44] to examine the potential mediating role of food integrity knowledge in the relation between food integrity climate and food integrity behavior. Hypothesis 3A states that knowledge might explain the relation between food integrity climate and food integrity behavior. The first step of the mediation analysis was already performed to test Hypothesis 1, and it was confirmed. There was a direct relation between food integrity climate and food integrity behavior (F (4, 102) = 19.27, *p* < 0.001). To test the second step of Baron and Kenny [44], namely, the effect of the independent variable on the mediator, a hierarchical linear regression analysis was performed, with knowledge as dependent variable. First, the control variables (company and conscientiousness) were included, for which a significant relation with knowledge was verified (F (4, 108) = 16.68, *p* < 0.001). Next, food integrity climate was added to the model and was found to have a positive significant association with knowledge (F (1, 107) = 5.57, *p* = 0.02). This implies that a strong food integrity climate is related to a higher knowledge about food integrity (β = 0.20, *p* = 0.02) (Table 6). This finding is also in line with the results from the bivariate correlations which showed that food integrity climate and food integrity knowledge were significantly positively correlated (r = 0.40, *p* < 0.01) (Table 2). In the third step, the relation between food integrity climate and food integrity behavior was examined while also controlling for food integrity knowledge. Results showed a significant association between knowledge and food integrity behavior (F (1, 100) = 21.57, *p* < 0.001), indicating that the more employees believe they hold knowledge on food integrity, the better they perceive their food integrity behavior. Importantly, the initial strong statistically significant relation between food integrity climate and food integrity behavior (β = 0.38, *p* < 0.001) weakened but remained statistical significant (β = 0.24, *p* = 0.002). Therefore, Hypothesis 3A could only partially be confirmed. This implies that the positive relation between food integrity climate and food integrity behavior can partially be explained by employees’ food integrity knowledge, meaning that a strong food integrity climate is associated with high levels of knowledge on food integrity among employees, which, in turn, is associated with a better food integrity behavior. 

#### 4.2.5. Hypothesis 3B: Food Integrity Motivation as a Mediator

A mediation analysis was performed following the procedure of Baron and Kenny [44] to also examine the potential mediating role of food integrity motivation in the relation between food integrity climate and food integrity behavior. Hypothesis 3B assumes that food integrity motivation mediates the relation between food integrity climate and food integrity behavior. The first hierarchical linear regression analysis is equivalent to Hypothesis 1 and was confirmed. There was a direct relation between food integrity climate and food integrity behavior (F (4, 102) = 19.27, *p* < 0.001). In the second step, the effect of the independent variable was analyzed against the mediator. A significant association between food integrity climate and motivation was found (β = 0.39, *p* < 0.01) while checking for the control variables (company and conscientiousness) (F (1, 107) = 25.99, *p* < 0.001) (Table 6). This is in line with the results from the bivariate analysis which showed that food integrity climate and food integrity motivation correlate significantly (r = 0.58, *p* < 0.01) (Table 2). In the third step, the relation between food integrity climate and food integrity behavior was examined, while also controlling for food integrity motivation (F (1, 100) = 25.18, *p* < 0.001). Results showed a significant positive association between motivation and food integrity behavior (β = 0.44, *p* < 0.001), meaning that the more employees are motivated to act with integrity, the better they perceive their food integrity behavior. Importantly, the initial strong statistically significant relation between food integrity climate and food integrity behavior dropped and even became statistically insignificant (β = 0.14, *p* = 0.07). Therefore, Hypothesis 3B could be fully confirmed. This implies that the positive relation between food integrity climate and food integrity behavior can fully be explained by employees’ food integrity motivation, indicating that a strong food integrity climate is associated with high levels of motivation to act with integrity, which, in turn, is associated with a better food integrity behavior. 

## 5. Discussion

Given the compelling need to prevent food fraud within the international food supply chain and the current lack of research on food integrity, in this paper, the concept of food integrity climate [9] was examined in light of employees’ behavior, to verify the extent to which the individual or human dimension plays a role in determining the organizational climate in terms of food integrity (or food fraud) and to explore whether a relation exists between a company’s food integrity climate and employees’ food integrity behavior. To this purpose, the construct of food integrity behavior was introduced and defined, and the conceptual model of food integrity climate in relation to food integrity behavior was elaborated along with study hypotheses. The proposed model took account of the potential moderating role of employees’ psychological well-being (i.e., burnout and job stress). Moreover, building upon work psychological theory, two mediating variables were proposed (i.e., knowledge and motivation) which both could explain how the prevailing food integrity climate might influence employees’ food integrity behavior. Data were collected in a convenience sample of 4 Belgian food companies with a total of 118 participating employees through a self-assessment questionnaire. 

Firstly, based on the statistical results obtained, a high food integrity climate was found to be associated with good food integrity behavior of the employees, confirming Hypothesis 1. This outcome suggests that the food integrity behavior might be rooted in the perceived food integrity climate of an organization, and, therefore, setting a high food integrity climate that is well-perceived by the employees is critical in ensuring a positive food integrity behavior. This finding corroborates the results of the descriptive analysis, which showed that the four participating companies rank in the same order, both in their climate and in their behavior scores, suggesting that a strong food integrity climate likely is associated with positive food integrity behavior of employees. This also falls in line with previous research in food safety context, which demonstrated that the food safety climate is positively associated with employee behavior in terms of food safety and hygiene [8,45,46]. 

Specifically, when the defining components of food integrity behavior (i.e., compliance, participation and pro-organizational unethical behavior) were analyzed separately, it was found that compliance and participation were significantly positively related to food integrity climate. However, no statistically significant relation was found between food integrity climate and unethical pro-organizational behavior. A high food integrity climate is possibly more strongly related to food integrity related outcomes (i.e., employees’ food integrity knowledge and motivation, employees’ compliance and participation regarding food integrity) then to more generic outcomes such as employees’ unethical pro-organizational behavior. In fact, unethical pro-organizational behavior goes beyond and is broader than food integrity. In line with our study results, employees’ (un)ethical pro-organizational conduct at work is likely more dependent on particular individual personality traits (i.e., conscientiousness) or beliefs (i.e., ethical beliefs) and less on the prevailing food integrity climate.

In this respect, an interesting effect of conscientiousness in the relation between food integrity climate and food integrity behavior was demonstrated, meaning that employees who indicated themselves to be conscientious tend to report better food integrity behavior. This result is in line with our expectations and is empirically supported by the literature. In fact, a significant relation was already proven to exist between employees’ conscientiousness and their behavior in the context of food safety [8]. A person who is conscientious tends to follow prescribed standards and rules [38]. Conscientiousness, therefore, might possess a predictive value for enhancing food integrity behavior besides food integrity climate. Possibly, both organizational (e.g., food integrity climate) and individual (e.g., personality) factors influence employees’ work behavior and should be considered when studying prevention solutions for food fraud in food companies.

Secondly, no statistical evidence was discovered for job stress and burnout as influencing or moderating factors in the relation between food integrity climate and food integrity behavior. Therefore, in line with the research by De Boeck et al. [8], Hypotheses 2A and 2B were not confirmed. Possible explanations for the statistical absence of these interactions may be related to the fact that job stress was assessed by means of a single item in the questionnaire, and burnout was surveyed with only three items, which could have been too general. Even the sample size was relatively small (*n* = 118 participants). Although no statistically significant relation was found for the moderating role of burnout and job stress in the relation between climate and behavior, this does not mean that the psychosocial well-being of employees in food companies is unimportant. Results from the bivariate correlations show, in fact, that high levels of burnout are significantly associated with a worse food integrity climate, less knowledge of food integrity, less motivation to act with integrity, less conscientiousness and, ultimately, worse food integrity behavior (Table 2). There is, therefore, a shared responsibility in the organization to pay attention to employees’ psychological well-being and to reduce or prevent burnout and job stress, which both can have negative effects on work performance [47]. This is crucial for both the success of the organization and the employees individually. Management should address these risk factors by implementing stress management interventions [48,49].

Thirdly, statistical results demonstrated that food integrity knowledge partially explains or mediates the relation between food integrity climate and food integrity behavior, which partly confirms Hypothesis 3A and is in line with the findings of De Boeck et al. [8] and Griffin and Neal [34] where the same results were found, respectively, in the contexts of food safety and healthcare. Furthermore, it was found that food integrity motivation fully explains or mediates the relation between food integrity climate and food integrity behavior, conforming Hypothesis 3B, even though, in De Boeck et al. [8], only a partial mediation of motivation was demonstrated in the context of food safety. Together, these results suggest that the prevailing food integrity climate in a food company might influence employees’ food integrity behavior through influencing employees’ knowledge and motivation regarding food integrity. In fact, previous work-related psychological research in a variety of industrial sectors demonstrated that work behavior and work performance are significantly influenced by employees’ motivation and knowledge [47]. In summary, results of this study demonstrate that companies’ food integrity climates possess both a direct and an indirect effect on employees’ food integrity behavior.

To prevent food fraud and promote food integrity behavior in a food company, it is recommended, according to the study results, to develop and shape a mature food integrity climate within the organization, to enhance food integrity knowledge and to boost food integrity motivation among employees. Companies should optimize and invest in education and training on food integrity. However, it is not sufficient to focus on knowledge transfer. Motivating employees is also essential. Leaders should understand the importance of motivation because organizational success depends on it [50]. Food organizations should establish procedures to facilitate training that boost knowledge, motivation, compliance and participation regarding food integrity [51]. Finally, regular evaluations of employees’ food integrity knowledge and motivation should become an integral part of organizational programs. Previous research already showed the importance of employees’ knowledge and self-motivation to perform safe food processing practices [52,53] and to ensure efficient team functioning [54]. Neal, Binkley and Henroid [55] also demonstrated that if the management is committed to creating a work environment that encourages good food safety behavior and culture, food service operators may be able to reduce the risk of foodborne illness outbreaks.

Although comparisons were made between previous similar research on food safety and the present study on food integrity, it must be remembered that food integrity is more comprehensive than food safety, since it includes even the aspects of food quality, authenticity, defense and all the practical and ethical processes of food production and distribution occurring along the food supply chain [9,56]. Measuring the food integrity climate can help management to identify potential system failures. Instead of basing actions on mistakes that occurred in the past, a shift in thinking is needed. Organizations should measure and act upon employees’ food integrity perceptions in order to work preventively and systematically [57]. Specifically, food companies should evolve from a compliance-based organizational food safety climate to an ethically strong organizational climate that focuses on the more comprehensive concept of integrity [58].

## 6. Limitations and Future Research

When interpreting these results, it is important to take a number of study limitations into account. The use of a cross-sectional design represents a first limitation, as it can have drawbacks. On the one hand, it is difficult to draw causal conclusions [59] and, on the other hand, the answers obtained represent snapshots [60]. Respondents may have a recent situation in mind when scoring the food integrity climate, and this could bias their answers. Although strong relations were identified in this study between food integrity climate and food integrity behavior, further longitudinal research is needed to uncover possible causal or reversed effects. A second limitation is that the questionnaire is self-reporting, which can lead to social desirability, among other things [61]. Participants could be ashamed of the fact that they have behaved against food integrity and may be tempted to score their food integrity behavior higher than it actually is. However, Spector [62] argues that the social desirability when using self-assessment measurement tools causes only minor variance. Moreover, in this study, social desirability was minimized by ensuring anonymity and confidentiality to participants. Finally, the relatively small sample size (*n* = 118 participants) may have also been a disadvantage, as this might have caused less statistical power and missing of relations between study variables. Further research could overcome this by replicating the study in a larger sample size and triangulating the results with qualitative methods to obtain deeper insights and further reduce response biases. 

Future research is recommended in the context of food integrity behavior exploring the role of other human factors or employees’ personal characteristics (e.g., personality, social norms, values, moral awareness, trust) as well as of work and organizational characteristics (e.g., performance, ethical leadership, work demands, work resources, food fraud management systems), since it can be expected that the context and characteristics of an organization influence food integrity behavior [63]. Finally, the role of social factors, such as social relationships or psychological processes within teams, could also be studied, since these were proven to have an influence on the climate and employees’ behavior within organizations [64,65].

## 7. Conclusions

Based on the statistical analysis on the survey data collected within four Belgian food companies (*n* = 118 participants), it can be concluded that: (1) the companies’ food integrity climate is positively related to the employees’ food integrity behavior, both directly and indirectly, (2) food integrity knowledge is a partial mediator in the relation between food integrity climate and food integrity behavior and (3) food integrity motivation is a full mediator in the relation between food integrity climate and food integrity behavior.

These results demonstrate the importance of considering human factors in the context of food integrity and suggest that merely applying technical standards and procedures may not be sufficient to achieve an optimal food integrity climate or prevent food fraud. 

## Figures and Tables

**Figure 1 foods-11-02657-f001:**
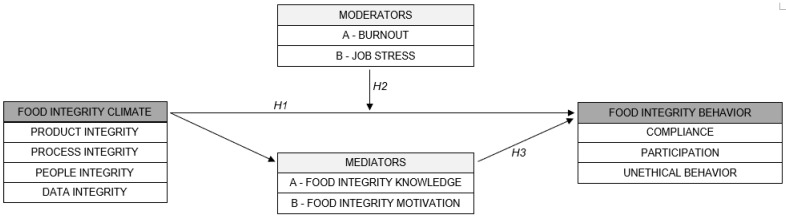
Conceptual model of the relation between food integrity climate and food integrity behavior. The food integrity climate as well as the moderators and mediators represent the continuous independent variables, while food integrity behavior is the continuous dependent variable of the study. It is assumed that food integrity climate is positively related to food integrity behavior (Hypothesis 1 (*H1*)), that burnout and job stress moderate the relation between food integrity climate and food integrity behavior (Hypothesis 2A and 2B (*H2*)) and that food integrity knowledge and motivation mediate the relation between food integrity climate and food integrity behavior (Hypothesis 3A and 3B (*H3*)).

**Table 1 foods-11-02657-t001:** Demographic analysis of the sample (*n* = 118 participants) and relation between categorical independent variables and dependent variable (food integrity behavior) (** *p* < 0.01, *** *p* < 0.001).

Variable	Percentage	Mean	Standard Deviation	F-Value
**Company**				5.76 **
A	11.9	63.23	5.51	
B	28	55.28	7.47	
C	29.7	58.22	5.96	
D	30.5	59.66	5.06	
**Age**				1.23
<26 years old	8.5	57.10	3.99	
Between 26–30 years old	11.9	56.21	5.86	
Between 31–40 years old	18.6	60.63	5.74	
Between 41–50 years old	28.8	59.39	7.11	
Between 51–60 years old	24.6	57.30	7.44	
>60 years old	7.6	59.75	4.59	
**Seniority**				0.95
<1 year	7.6	59.86	8.40	
Between 1–5 years	27.1	58.10	6.21	
Between 6–10 years	14.4	59.88	6.38	
Between 11–15 years	17.8	59.58	8.32	
Between 16–20 years	8.5	54.90	6.23	
>20 years	24.6	58.32	4.83	
**Function**				1.61
Management	10.2	61.50	6.43	
Daily contact with food	67.8	57.92	6.65	
No direct contact with food	22	58.72	5.83	
**Contract**				0.16
Permanent contract	85.1	58.43	6.29	
Temporary contract	14.9	59.38	9.09	
**Conscientiousness**				7.40 ***

**Table 2 foods-11-02657-t002:** Univariate analysis of continuous study variables and bivariate correlation matrix of continuous study variables (*n* = 118 participants). Internal Consistency Reliability (α) is shown in the diagonal (** *p* < 0.01) (/ = out of) (SD = standard deviation). Food integrity behavior was found to have a significant positive relation with conscientiousness, food integrity climate, food integrity knowledge and food integrity motivation, a significant negative relation with burnout and no significant relation with job stress.

Variable	Mean	SD	1.	2.	3.	4.	5.	6.	7.
1. Conscientiousness	20.23/25	2.58	(0.67)						
2. Food Integrity Climate	78.08/100	10.64	0.35 **	(0.91)					
3. Burnout	8.06/21	3.70	−0.31 **	−0.44 **	(0.62)				
4. Job stress	3.37/7	1.65	−0.05	−0.32 **	0.49 **	-			
5. Knowledge	25.55/30	3.04	0.58 **	0.40 **	−0.36 **	−0.03	(0.82)		
6. Motivation	30.59/35	3.37	0.56 **	0.58 **	−0.37 **	−0.08	0.72 **	(0.83)	
7. Food Integrity Behavior	58.50/70	6.48	0.55 **	0.54 **	−0.42 **	−0.04	0.64 **	0.71 **	(0.84)

**Table 3 foods-11-02657-t003:** Hierarchical linear regression of food integrity behavior and its three subscales (i.e., compliance, participation and unethical behavior) on control variables and food integrity climate (*n* = 118 participants). Standardized regression coefficients for the respective regression steps are reported (* *p* < 0.05, ** *p* < 0.01, *** *p* < 0.001; Company A is reference category). Hypothesis 1 was confirmed, since a positive relation between the perceived food integrity climate and food integrity behavior was demonstrated.

Predictor	Food Integrity Behavior		Compliance		Participation		Unethical Behavior	
	Model 1	Model 2	Model 1	Model 2	Model 1	Model 2	Model 1	Model 2
**1. Control Variables**								
Company								
B	−0.39 **	−0.31 **	−0.32 *	−0.21	−0.42 **	−0.35 **	−0.15	−0.10
C	−0.32 **	−0.24 *	−0.25 *	−0.14	−0.47 ***	−0.40 *	−0.07	−0.03
D	−0.05	−0.05	−0.14	−0.12	−0.26 *	−0.26 *	0.17	0.18
Conscientiousness	0.57 ***	0.46 ***	0.54 ***	0.40 ***	0.48 ***	0.39 ***	0.34 ***	0.27 **
**2. Food Integrity Climate**		0.31 ***		0.39 ***		0.25 **		0.18
R^2^	0.43 ***	0.51 ***	0.36 ***	0.48 ***	0.34 ***	0.39 **	0.17 **	0.19
Adjusted R^2^	0.41 ***	0.49 ***	0.34 ***	0.46 ***	0.32 ***	0.36 **	0.14 **	0.15
ΔR^2^	0.43 ***	0.08 ***	0.36 ***	0.12 ***	0.34 ***	0.05 **	0.17 **	0.03

**Table 4 foods-11-02657-t004:** Hierarchical linear regression of the moderator job stress (*n* = 118 participants). Standardized regression coefficients for the respective regression steps are reported. Conscientiousness, food integrity climate and job stress are centered around the mean (** *p* < 0.01, *** *p* < 0.001; Company A is reference category). Hypothesis 2A was rejected, since job stress appeared not to have a moderating effect in the relation between food integrity climate and food integrity behavior.

Predictor	Food Integrity Behavior		
	Model 1	Model 2	Model 3
**1. Control Variables**			
Company			
B	−0.39 **	−0.31 **	−0.31 **
C	−0.32 **	−0.20	−0.22
D	−0.05	−0.02	−0.02
Conscientiousness	0.57 ***	0.45 ***	0.43 ***
**2. Food Integrity Climate**		0.36 ***	0.38 ***
Job stress		0.14	0.12
**3. Food Integrity Climate × Job stress**			−0.13
R^2^	0.43 ***	0.53 ***	0.54
Adjusted R^2^	0.41 ***	0.50 ***	0.51
ΔR^2^	0.43 ***	0.10 ***	0.02

**Table 5 foods-11-02657-t005:** Hierarchical linear regression of the moderator burnout (*n* = 118 participants). Standardized regression coefficients for the respective regression steps are reported. Conscientiousness, food integrity climate and burnout are centered around the mean (* *p* < 0.05, ** *p* < 0.01, *** *p* < 0.001; Company A is reference category). Hypothesis 2B was rejected, since burnout appeared not to have a moderating effect in the relation between food integrity climate and food integrity behavior.

Predictor	Food Integrity Behavior		
	Model 1	Model 2	Model 3
**1. Control Variables**			
Company			
B	−0.39 **	−0.29 *	−0.27 *
C	−0.32 *	−0.26 *	−0.25 *
D	−0.05	−0.07	−0.07
Conscientiousness	0.57 ***	0.43 ***	0.44 ***
**2. Food Integrity Climate**		0.27 **	0.28 **
Burnout		−0.13	−0.11
**3. Food Integrity Climate × Burnout**			0.06
R^2^	0.42 ***	0.52 ***	0.52
Adjusted R^2^	0.40 ***	0.49 ***	0.49
ΔR^2^	0.43 ***	0.10 ***	0

**Table 6 foods-11-02657-t006:** Hierarchical linear regression of food integrity knowledge and food integrity motivation on control variables and food integrity climate (step 2 of the mediation analysis) (*n* = 118 participants). Standardized regression coefficients for the respective regression steps are reported. Conscientiousness and food integrity climate are centered around the mean (* *p* < 0.05, ** *p* < 0.01; Company A is reference category). Hypotheses 3A and 3B were confirmed, since the relation between food integrity climate and food integrity behavior was partially mediated by food integrity knowledge and fully mediated by food integrity motivation.

Predictor	Food Integrity Knowledge		Food Integrity Motivation	
	Model 1	Model 2	Model 1	Model 2
**1. Control Variables**				
Company				
B	−0.33 *	−0.27 *	−0.34 *	−0.27 *
C	−0.30 *	−0.25 *	−0.30 *	−0.18
D	−0.27 *	−0.27 *	−0.17	−0.15
Conscientiousness	0.55 **	0.48 **	0.55 **	0.40 **
**2. Food Integrity Climate**		0.20 *		0.39 **
R^2^	0.38 **	0.41 *	0.37 **	0.49 **
Adjusted R^2^	0.36 **	0.39 *	0.35 **	0.47 **
ΔR^2^	0.38 **	0.03 *	0.37 **	0.12 **

## Data Availability

The data that support the findings of this study are available from the corresponding author upon reasonable request.

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
