# Peer review of "An Exploratory Study on the Relation between Companies’ Food Integrity Climate and Employees’ Food Integrity Behavior in Food Businesses"

_foods, 2022, doi:10.3390/foods11172657_

Round 1

Reviewer 1 Report

Dear Authors,

The manuscript (foods-1889765) presented for review is interesting.

 Authors, Please note and address the following comments:

Title: In my opinion, it should be added to the title that is "a preliminary study".

Keywords: In my opinion, keywords should be rethought. It is difficult to understand the topic of the article based on the keywords.

Introduction: The background of this study is poor.

Results

The information in chapter 4.1 is the characteristics of the studied group and should be in the Materials chapter.

In my opinion, the authors took into account too few factors that may affect the integrity of employees regarding food integrity behavior in the food business.

I wonder if the selection of the sample (companies) is correct.

Most companies operate within the limits of food law and are honest, and the conscientiousness of employees is related to the nature of the person/employee.

Limitation: I propose to separate the limitations and strengths of these results.

Technical Notes

References: References are not cited according to journal rules. Publications from MDPI provide information on how to properly cite. Authors may also find this information in the authors' guide.

 The font in the survey questions differs from the one in the article.

 Despite my comments, I am pleased to recommend this manuscript for publication, but it should be shortened and better explain. I believe it addresses an important area of research in an international context.

 Reviewer

Reviewer 2 Report

Comment: Thank you for the opportunity to review the manuscript entitled “Exploring the relation between companies’ food integrity climate and employees’ food integrity behavior in food businesses” submitted for publication to Foods. The manuscript explores the relationship between the organizational food integrity climate and the employees’ food integrity.

Abstract. The abstract is clear and comprehensive. The authors define the purpose of the research, as well as the materials and methods and the main outcomes of the research. 

General comment. References should be modified according to the Instruction for Authors provided by MDPI. 

Introduction. The authors can briefly introduce some previous research related to the topic, as to enlarge the theoretical background of the research and highlight the originality/novelty of the present research. 

Conceptual research model. Figure 1 should be included into the text, soon after Figure 1 is cited, not at the beginning of the section “Conceptual research model”. Although the authors provide definition for each variable include in the research model, and also if the authors have included clear hypotheses, Figure 1 is not clear and does not provide useful information related to the hypothesis development. At present, Figure 1 does not summarize the research model in a clear manner. It is not clear the starting point of the research method, as well as the different relations between moderators and mediators. 

Materials and methods. Lines 300-301 define a “literature review”, which was at the basis of the development of the 61 questions. Could you please define the literature review process, and the articles useful to develop 61 questions? Moreover, I suppose that 61 questions are a great number of questions, could you please provide examples of successful articles, which adopted a great number of questions?

Line 325. Why “fourteen” is written in letters and not in numbers?

Why some variables (e.g., food integrity behavior, food integrity knowledge, food integrity motivation) have been calculated on the basis of a five-point Likert scale (line 326), whereas other variables (e.g., job stress, burnout) on the basis of a seven-point Likert scale (line 340)?

Results. Results are provided in a clear and comprehensive manner. The authors first provide descriptive statistics results, then specific results related to the investigated variables. Also, the authors discuss point-by-point each developed hypothesis. Results are significant. 

Discussion. Discussions are relevant, and the authors develop theoretical and managerial implications. 

Appendix. The authors provide the entire questionnaire, which help readers understanding the research and the analysis process. 

Reviewer 3 Report

Food fraud is the deliberate and intentional substitution, addition, tampering or misrepresentation of food products or ingredients, which causes damage to the consumer not only from a hygienic-sanitary point of view, but also from an economic point of view. Ensuring the integrity of food throughout the food supply chain, from farm to fork, means ensuring that the food product is not subject to fraud.

This work aims to evaluate the relationship between the organizational climate of food integrity and the food integrity behavior of employees to understand the role of the individual or psychological dimension in food integrity.

The paper deals with an important issue, in fact all operators in the food chain, whatever the point on which they intervene, have the duty to preserve the integrity of food and to supervise the safety of operations.

Overall

The structure of the manuscript is correct. Among other things, the authors also included an appendix with the questionnaires administered to the participants. The references are present in sufficient numbers.

Although further research is needed, the processing of the data obtained allows to provide useful information on the relationship between the climate of food integrity and food integrity behavior.

However, some changes are required, as follows:

The Conclusions and future research section should not contain references. Therefore, I suggest the authors reshape this section, moving the part with references to the discussion section and reporting only future research in the conclusions.

According to the Instructions for authors, the references should be numbered in order of appearance and indicated by a numeral or numerals in square brackets.
